# Drugs Commonly Applied to Kidney Patients May Compromise Renal Tubular Uremic Toxins Excretion

**DOI:** 10.3390/toxins12060391

**Published:** 2020-06-12

**Authors:** Silvia M. Mihaila, João Faria, Maurice F. J. Stefens, Dimitrios Stamatialis, Marianne C. Verhaar, Karin G. F. Gerritsen, Rosalinde Masereeuw

**Affiliations:** 1Division of Pharmacology, Utrecht Institute for Pharmaceutical Sciences, Utrecht University, 3854 CG Utrecht, The Netherlands; s.mihaila@uu.nl (S.M.M.); M.C.Verhaar@umcutrecht.nl (M.C.V.); K.G.F.Gerritsen@umcutrecht.nl (K.G.F.G.); 2Department of Nephrology and Hypertension, University Medical Center Utrecht, 3582 CX Utrecht, The Netherlands; j.p.ferreirafaria@uu.nl (J.F.); m.f.j.stefens@students.uu.nl (M.F.J.S.); 3(Bio)artificial Organs, Department of Biomaterials Science and Technology, University of Twente, 7522 LW Enschede, The Netherlands; d.stamatialis@utwente.nl

**Keywords:** protein-bound uremic toxins, chronic kidney disease management, drug-toxin interaction, OAT1-mediated transport

## Abstract

In chronic kidney disease (CKD), the secretion of uremic toxins is compromised leading to their accumulation in blood, which contributes to uremic complications, in particular cardiovascular disease. Organic anion transporters (OATs) are involved in the tubular secretion of protein-bound uremic toxins (PBUTs). However, OATs also handle a wide range of drugs, including those used for treatment of cardiovascular complications and their interaction with PBUTs is unknown. The aim of this study was to investigate the interaction between commonly prescribed drugs in CKD and endogenous PBUTs with respect to OAT1-mediated uptake. We exposed a unique conditionally immortalized proximal tubule cell line (ciPTEC) equipped with OAT1 to a panel of selected drugs, including angiotensin-converting enzyme inhibitors (ACEIs: captopril, enalaprilate, lisinopril), angiotensin receptor blockers (ARBs: losartan and valsartan), furosemide and statins (pravastatin and simvastatin), and evaluated the drug-interactions using an OAT1-mediated fluorescein assay. We show that selected ARBs and furosemide significantly reduced fluorescein uptake, with the highest potency for ARBs. This was exaggerated in presence of some PBUTs. Selected ACEIs and statins had either no or a slight effect at supratherapeutic concentrations on OAT1-mediated fluorescein uptake. In conclusion, we demonstrate that PBUTs may compete with co-administrated drugs commonly used in CKD management for renal OAT1 mediated secretion, thus potentially compromising the residual renal function.

## 1. Introduction

Chronic kidney disease (CKD) is a worldwide public health problem associated with considerable prevalence of comorbidities, impaired quality of life and premature mortality [1]. In patients with advanced CKD, uremic solutes accumulate due to impaired renal clearance [2]. Many of them are considered uremic toxins (UTs) and are believed to contribute to the uremic syndrome, a generalized organ dysfunction occurring in CKD [3,4,5,6,7,8,9]. In particular, protein-bound uremic toxins (PBUTs) were shown to exert toxic effects or disrupt key signaling and metabolic pathways, including those controlled by a complex network of solute carrier (SLC) and ATP-binding cassette (ABC) transporters and drug-metabolizing enzymes (DMEs), many of which are critical for drug absorption, distribution, metabolism and elimination (ADME) [10,11]. As CKD progresses, drug–metabolite interactions involving transporters can become more deleterious, potentially contributing to new and increased drug toxicities [7,12,13,14,15].

CKD patients are routinely treated with many drugs that require transporters and DMEs for their disposition [11,16]. The daily medication burden in kidney patients is one of the highest reported to date in any chronic disease state [17], and drugs are prescribed mainly at alleviating the metabolic, endocrine and cardiovascular complications in renal insufficiency [18,19,20,21]. Antihypertensive drugs (e.g., renin–angiotensin–aldosterone system inhibitors, such as angiotensin-converting enzyme inhibitors (ACEIs) or angiotensin receptor blockers (ARBs), and diuretics) and cholesterol-lowering drugs (e.g., statins) are frequently prescribed medications for cardiovascular risk management in the CKD population [20,22,23,24,25,26,27,28]. With drug–drug interactions occurring even in patients without renal impairment, such interactions are likely more prevalent in CKD owing to the presence of high levels of uremic solutes that also compete with the administered drugs and with each other for transporters and DMEs [7].

In renal tissue, the key mediators of transepithelial transport of many endogenous metabolites are the organic anion transporters (OATs), which play a central role in the cellular uptake of uremic retention solutes as a first step in their excretion [11,29,30]. Amongst the OATs, OAT1 (*SLC22A6*) and OAT3 (*SLC22A8*) are abundantly expressed at the basolateral membrane of proximal tubule cells, and appear to be the most important transporters involved in the renal uptake of endogenous metabolites, including PBUTs (e.g., indoxyl sulfate, p-cresylsulfate, kynurenic acid, etc.), with OAT1 dominating over OAT3 with respect to PBUT uptake [10,29,31,32], drugs (probenecid, methotrexate, adefovir, indomethacin) and other exogeneous toxins [33]. The interactions between PBUTs and commonly prescribed drugs are currently unknown.

The aim of this study was to investigate the pharmacokinetic interactions between commonly prescribed drugs in CKD management (ACEIs, ARBs, statins and furosemide) and PBUTs with respect to OAT1-mediated uptake in an in vitro setting. First, the inhibitory potency of the selected PBUTs and drugs on OAT1 activity was evaluated. For this, we exposed a human conditionally immortalized proximal tubule cells line (ciPTEC) expressing OAT1 (ciPTEC-OAT1) previously developed and characterized [34,35,36,37], to variable concentrations of PBUTs or drugs and evaluated fluorescein uptake, a known substrate for OAT1 [38]. Subsequently, we explored the effect of PBUTs presence, at concentrations similar to those in plasma of CKD patients, on the drug profiles, again, by studying OAT1-mediated fluorescein uptake.

## 2. Results

### 2.1. Protein-Bound Uremic Toxins Reduce OAT1-Mediated Uptake at Clinically Relevant Concentrations

First, the effect of PBUTs on OAT1-mediated uptake of fluorescein by ciPTEC-OAT1 was evaluated. Fluorescein, an OAT1 model substrate, was used to evaluate the transporter activity as described earlier [36]. To confirm the OAT1 specificity of fluorescein uptake, we have co-incubated fluorescein with probenecid, a well-known OAT1 inhibitor [39,40], which resulted in a fluorescein uptake of 24.9 ± 3.2% of the fluorescein alone values. This confirms the stable OAT1 activity on our cell line, as shown in previous studies [36]. Next, the decrease in fluorescein uptake in the presence of PBUTs was evaluated and was considered to occur via competition for the transporter (Figure 1A). The selection of the uremic toxins in the UTox mixture was based on their reported proximal tubule-mediated urinary secretion profile and association with CKD progression [36,41,42]. Indoxyl sulfate (IS), kynurenic acid (KA) and p-cresylsulfate (PCS) reduced fluorescein uptake at clinically relevant concentrations (viz. 110 µM, 1 µM and 125 µM, respectively). The concomitant exposure to a mixture of eight uremic toxins (UTox; for composition see Appendix A) further reduced fluorescein uptake (Figure 1B, to 34.4 ± 8.3%, *p* < 0.0001 when compared to control).

The reduction of fluorescein uptake in the presence of probenecid (500 µM) was in agreement with our previous findings [36]. Additionally, the transport kinetics of OAT1-mediated fluorescein uptake was investigated by studying the concentration-dependent uptake of the substrate in the presence of selected uremic toxins. The fluorescein uptake by ciPTEC-OAT1 followed Michaelis-Menten kinetics from which kinetic parameters (Km and Vmax values) were determined (Appendix A and Appendix A).

### 2.2. Commonly Prescribed Drugs in CKD Management Reduce OAT1-Mediated Uptake

To evaluate the role of OAT1 in the disposition of commonly prescribed drugs in CKD management, a panel of nine drugs selected based on their use within our hospital (ACEIs: captopril, enalaprilate and lisinopril; ARBs: losartan and valsartan; statins: pravastatin and simvastatin and diuretics: furosemide, Appendix A) was selected. Cimetidine (a histamine H2 receptor antagonist, H2RA) and a model drug for inhibition of the organic cation transport (OCT2, *SLC22A2*) was used to reflect the non-inhibition of OAT1 [43].

A concentration-dependent reduction of fluorescein uptake was observed for all drugs (Figure 2), with the most potent interactions found for ARBs and furosemide (approx. 50% reduction in fluorescein uptake at the highest therapeutic concentration) and statins (approx. 20% reduction in fluorescein uptake at the highest therapeutic concentration), while for ACEIs and cimetidine either no effect was measured or a decrease was observed only at the highest concentrations tested beyond the therapeutic range (Table 1). Notably, at the highest concentrations of the ARBs and furosemide, the reduction in fluorescein uptake reached values similar to those obtained for probenecid, suggesting a (nearly) complete inhibition of the transporter. In the case of statins, at the highest concentration, the reduction in fluorescein was modest, an indication of partial inhibition of the transporter’s activity. Thus, our data suggest that the selected ARBs, statins and furosemide act as transport inhibitors of OAT1-mediated uptake.

### 2.3. Commonly Prescribed Drugs in CKD in Combination with PBUTs Further Reduce OAT1-Mediated Uptake

Next, a dual competitive inhibition experiment with drugs and PBUTs on OAT1-mediated fluorescein uptake was performed, leaving out the ACEIs and cimetidine because of their suprapharmacological, clinically irrelevant interactions. To this end, mature monolayers of ciPTEC-OAT1 were co-incubated with the selected drugs at variable concentrations (as inhibitor 1) and PBUTs (IS, KA, PCS, UTox as inhibitor 2) at uremic concentrations, together with fluorescein.

In the presence of PBUTs, the inhibitory effect of the drugs on fluorescein uptake was maintained with a clear dose-response relationship. The IC_50_ values were affected by the presence of PBUTs as shown in Table 2. In the case of drug–IS co-incubation, fluorescein uptake was considerably reduced in the presence of IS at low drug concentrations as compared to incubation of the drug alone, suggesting a strong inhibition of the transporter primarily determined by IS. For the ARBs, fluorescein uptake showed an immediate progressive decrease with increasing drug concentrations already at low, therapeutic concentrations, while for furosemide and statins this decrease appeared only when co-incubation was performed at higher drug concentrations (Figure 3). In all cases, the maximum percentage reduction was in the same range as that of probenecid, suggesting a complete inhibition of the transporter.

With KA alone, only a slight reduction in fluorescein uptake was noticed. In co-incubation with drugs, the inhibitory potencies of the drugs were not aggravated (Figure 4, Table 2). In the case of ARBs, the co-incubation with KA resulted in an increase in fluorescein uptake when compared to drug alone.

With PCS, fluorescein uptake was strongly inhibited in the absence of drugs confirming the role of OAT1 in handling PCS [48]. During co-incubations with variable drug concentrations, a further reduction was noticed (Figure 5 and Table 2), similar to IS. Again, in the case of furosemide and ARBs, the additional reduction in fluorescein uptake was observed at concentrations within the therapeutic range of the drugs.

The exposure to the mixture of PBUTs (UTox) resulted in a strong reduction of fluorescein uptake (34.4 ± 8.3%). With the addition of drugs, a further decrease was observed at the lowest concentration of each drug (with values comparable to probenecid), with a more pronounced effect for losartan and valsartan. This drop remained stable with the increase of the drug concentration (Appendix A and Appendix A**)**.

## 3. Discussion

Our major findings are that PBUTs may directly interact with drugs commonly prescribed in CKD management for OAT1-mediated renal transport at concentrations found in uremic serum. These interactions could exert widespread and unpredictable effects in CKD patients that already have a high burden of co-existing diseases, poor health-related quality of life and are prescribed many medications [16,17]. CKD patients use multiple drugs, in particular for CKD-related complications such as hypertension, CVD and lipid disorders. There is a high variability in pharmacokinetics in CKD patients, which can be partly explained by drug–drug interactions (DDIs) resulting from this polypharmacy [49,50,51]. In addition, as CKD progresses and uremic toxins accumulate, drug–metabolite interactions involving transporters can become more prevalent and more deleterious, potentially contributing to new and/or increased drug toxicities [7]. Although not all drugs tested show a high renal clearance profile (Appendix A), we consider the findings still clinically relevant. In CKD, pharmacokinetics can be severely altered not only due to renal dysfunction but also affecting drug absorption and metabolism [52]. In addition, DDIs can take place on protein binding.

The potential interaction of drugs and metabolites with the renal organic ion secretion system (OAT1 and OAT3, organic cation transporter 2 (OCT2), and multidrug and toxin extrusion (MATE)) has been acknowledged by the US Food and Drug Administration (FDA). They released regulatory guidelines to study the contribution of (renal) transporters in disposition of new pharmaceutical entities to identify and understand potential DDIs and (drug-induced) nephrotoxicity including evaluation of interaction with the transporters in vitro [53]. Noteworthy, drugs for which the kidney is not the main route of excretion (losartan, valsartan, simvastatin), but have an inhibitory effect towards OATs, could contribute to the reduction of PBUTs clearance, and thus, to the progression of the disease. For this purpose, human models with a high predictive capacity for renal drug handling are being used [54,55,56]. The human-derived proximal tubule epithelial cell model, ciPTEC-OAT1, developed by us [34,35,36,37], was shown to be a robust in vitro model to study drug interactions. We here demonstrate the potency of commonly prescribed drugs in the management of CKD comorbidities (ACEIs, ARBs, statins and diuretics) [24,27,57,58] to inhibit basolateral OAT1-mediated uptake of fluorescein at clinically relevant concentrations.

The potential involvement of OAT1 in the handling of the drugs tested had been previously reported [56], but not in the context of uremia. Sato et al. [58] reported comparable IC_50_ values for losartan and valsartan (12 and 16 µM, respectively; compared with 8.6 ± 2.5 µM and 11.5 ± 3.5 µM reported here), based on the uptake of uric acid by OAT1-expressing Flp-HEK293 cells. For statins, exposure to our in vitro cell model revealed higher affinities towards OAT1 than those reported in the literature (pravastatin: 23.2 ± 8.3 µM vs. 408 ± 55 µM reported by Taketa et al. [47] and simvastatin: 21.3 ± 3.8 µM vs. 73.7 ± 6.6 µM reported in the same study). However, these experiments were performed in OAT1-expressing muscle cells and not in renal proximal tubule cells. With regard to ACEIs, we show that human OAT1 is inhibited by captopril and enalaprilate (albeit at high concentrations with no therapeutic relevance), but not by lisinopril, similar as reported for mouse Oat11 [59,60]. Therefore, we did not continue with further evaluation of these drugs in combination with uremic toxins. Further, previous studies performed by our group showed a strong inhibitory potential of furosemide, in contrast with the slight effect of cimetidine, but in line with the data presented here [36]. Noteworthy, some of the tested drugs were reported to interact with OAT3 as well [56,58].

The accumulation of PBUTs due to decreased renal excretion and gut dysbiosis is associated with several comorbidities and altered drug metabolism [61,62,63], thus further contributing to the progression of renal disease. Here, three PBUTs (IS, KA and PCS) were selected for drug interaction studies using the ciPTEC-OAT1 model, as it was previously reported that these toxins interact with OAT1 and have been associated with CKD progression and its related complications [37,48,64,65]. Competitive inhibition of drug transporters and the concomitant alteration in pharmacokinetics may result from elevated levels of multiple retention solutes within the serum [29,63]. To reflect this complexity, we also prepared a mixture of the most relevant PBUTs (UTox), at uremic concentrations taken from the EUTox Uremic Solutes Database [42]. Of note, the variability of PBUTs levels among individuals is large [66], thus the PBUTs concentrations of the UTox were chosen within the average levels. While confirming the inhibitory potential of the individual toxins (IS, KA, PCS), we showed a significantly stronger inhibition when cells were exposed to the UTox, but not enough to reach the inhibitory potential of probenecid, which we consider as reference inhibitor of OAT1. When co-incubating with selected drugs, a further decrease in fluorescein uptake was observed, reaching similar levels as those obtained with probenecid, suggesting a complete saturation of the transporter via competitive inhibition.

The simultaneous incubation of individual toxins at uremic concentrations (IS-110 µM, KA-1 µM and PCS-125 µM) with the selected drugs resulted in a significant alteration of the inhibition potential of drugs on OAT1-mediated fluorescein uptake, when compared to the toxins alone. This indicates that the transporter is susceptible to modifications of the microenvironmental composition. In the case of IS- and PCS-drug co-incubation, at the lowest concentration of the drugs tested, the inhibitory potential was similar to that of toxin alone, suggesting that the competitive inhibition was mainly toxin-dominated. With increasing drug concentrations, which in case of furosemide, losartan and valsartan was still within their therapeutic window, a significant inhibition occurred, which is further evidence of OAT1-mediated handling of the drugs, concomitant with the PBUTs (Appendix A). Of note, the concentrations of IS and PCS were higher than their IC_50_ values (IS IC_50_ = 25 ± 4 µM and PCS IC_50_ = 79 ± 14 µM) reported for the ciPTEC-OAT1 system [37]. KA on the other hand was studied here at a concentration lower than its reported IC_50_ (IC_50_ = 6 ± 1 µM) [37], which explains the slight reduction in fluorescein uptake.

For KA-ARBs co-incubation, the inhibitory potential was lower than when the cells were exposed to the drug alone, despite the increase in drug concentration, an indication that KA compromises the uptake of ARBs in favor of fluorescein, suggesting that the toxin acts as a negative allosteric modulator, an effect previously documented for *α*7 nicotinic receptors [67]. Such effect was not observed for furosemide and statins, for which the drug-KA curves overlapped with those of the drugs alone.

This study has some limitations that should be addressed in future research. First, PBUTs in plasma are highly bound to plasma proteins (90–95% for IS [68] and PCS and 70% for KA [69]) and their renal excretion depends largely on active tubular secretion, which shifts the binding and powers the active secretion of the free fraction. The same holds true for the drugs tested in our study (Appendix A). Thus, future research will be needed to elucidate the contribution of protein binding in drug–toxin interactions with the transporters. Recently, we have demonstrated that protein binding positively affects the renal tubular clearance of uremic toxins (IS and KA) using the ciPTEC-OAT1 model [37,70]. As demonstrated by others, the binding capacity of albumin is diminished in CKD patients, most likely due to posttranslational modifications of albumin sites which could contribute to less efficient transport of uremic toxins by the renal tubular excretory machinery [70,71,72], thus resulting in further elevated plasma levels and their well-known consequences. Moreover, studies have revealed that interactions between drugs and uremic toxins could result in altered protein binding affinities [73,74,75,76].

Second, our study design relied on the use of flat monolayers of ciPTEC-OAT1. Herein, the intracellular accumulation of fluorescein was used as a measure of OAT1 activity. With the uptake transporters at the basolateral site, the proximal tubule cells possess a series of renal efflux transporters in the apical membrane responsible for the excretion of endo- and xenobiotics into the pro-urine. Although this 2D configuration cannot replicate the basolateral and apical compartments and does not sustain the monolayer polarization, it allows for high throughput screening of multiple drugs, toxins and their combination. A 3D configuration would permit the exposure of drugs and toxins at the basolateral side, with excretion at the apical side, as previously shown by us [37]. However, such systems are complex, time consuming and difficult to standardize [55].

Kidney failure not only alters the renal excretion processes, but also the non-renal disposition of drugs that are extensively metabolized by the liver. Their pharmacokinetics are often unpredictable, possibly due to alterations in the expression and activity of extra-renal DMEs and transporters [77]. Uremic toxins interfere with transcriptional activation and directly inhibit the activity of many members of the cytochrome P450 enzyme (CYP) family and SLC and ABC drug transporters. Recent evidence suggests that PBUTs can alter the hepatocyte mitochondrial function and the bile acid transport and synthesis [78]. Therefore, drug-PBUTs interactions in relation to drug handling by hepatocytes is primordial.

To summarize, the results of this study indicate a potential interaction of commonly prescribed drugs in CKD and PBUTs secretion, thus bringing another layer of complexity in the management of CKD. Our results show that the drug–toxin interactions with transporters lead to altered and heterogeneous uptake patterns, based on their affinity for the transporters within a uremic microenvironment. The immediate effect of these interactions could result in potential inhibition of uremic toxins or drug excretion, thus leading to their accumulation within the blood and systemic toxicity. Finally, this study contributes to the further advancement in understanding drug–toxin interactions and the alterations in drug pharmacokinetics that aid the appropriate dose adjustment in CKD patients.

## 4. Materials and Methods

### 4.1. Chemicals

A brief overview of the uremic toxins used within the study (indoxyl sulfate, indoxyl-β-d-glucoronide, indole-3-acetic acid, kynurenic acid, l-kynurenine, hippuric acid, p-cresylsulfate, p-cresylglucuronide) and tested drugs (ACEIs, ARBs, statins, furosemide and cimetidine) is available in Appendix A section (Appendix A, respectively). Chemicals were purchased from Sigma-Aldrich (Zwijndrecht, The Netherlands) unless stated otherwise. The uremic toxins p-cresylsulfate and p-cresylglucuronide were synthesized by the Institute for Molecules and Materials, Radboud University, Nijmegen, The Netherlands, as described earlier [37].

### 4.2. Cell Cultures

Conditionally immortalized proximal tubule epithelial cells, obtained from urine samples of healthy volunteers and overexpressing the organic anion transporter 1, (ciPTEC-OAT1) were cultured as described by Nieskens et al. [36]. Briefly, cells were cultured up to maximum 60 passages in Dulbecco’s Modified Eagle Medium/Nutrient Mixture F-12 (1:1 DMEM/F-12) (Lonza, Basel, Switzerland supplemented with 10% fetal calf serum (FCS) (Greiner Bio-One, Alphen aan den Rijn, The Netherlands), 5 μg/mL insulin, 5 μg/mL transferrin, 5 μg/mL selenium, 35 ng/mL hydrocortisone, 10 ng/mL epidermal growth factor and 40 pg/mL tri-iodothyronine to form a complete culture medium, without addition of antibiotics. Cells were cultured at 33 °C and 5% (*v*/*v*) CO_2_ to allow expansion and prior to the experiments seeded at a density of 63,000 cells/cm^2^. Subsequently, cells were grown 24 h at 33 °C, 5% (*v*/*v*) CO_2_ to allow adhesion and proliferation, then cultured for 7 days at 37 °C, 5% (*v*/*v*) CO_2_ for differentiation and maturation, refreshing the medium every other day. The temperature shift ensured the maturation of cells into fully differentiated epithelial cells able to form confluent monolayers.

### 4.3. Fluorescein Inhibition Assay

The potency of a panel of several commonly prescribed drugs in the CKD management to inhibit OAT1-mediated fluorescein uptake was investigated in ciPTEC-OAT1 mature monolayers using an inhibition assay, as previously described by Nieskens et al. [36]. CiPTEC-OAT1 cultured in 96-well plates were co-incubated with fluorescein (1 µM, unless otherwise stated) and the selected drugs (Appendix A) at different concentrations (nM-mM) prepared in Krebs-Henseleit Buffer supplemented with 10 mM HEPES (KHH buffer, pH 7.4) for 10 min at 37 °C. To confirm the activity of OAT1, probenecid (500 µM in KHH) was simultaneously incubated with fluorescein. Uptake arrest was performed by washing with ice-cold HBSS (Life Technologies Europe BV, Roskilde, Denmark), and then the cells were lysed by 100 µL 0.1 M NaOH for 10 min, at room temperature and under mild shaking. Intracellular fluorescence was detected using the Fluoroskan Ascent FL Spectrophotometer Labsystems (Life Technologies Europe BV, the Netherlands) microplate reader (), at excitation wavelength of 490 nm and emission wavelength of 518 nm. Background values were subtracted and normalized arbitrary fluorescence unit (AFU) data were converted into percentage (%). Incubation with fluorescein alone was assigned as 100% uptake. The reduction in fluorescein uptake in the presence of the inhibitor (drugs) was normalized to fluorescein uptake without the inhibitor.

### 4.4. Co-Exposure of ciPTEC-OAT1 to Selected PBUTs and Drugs

To confirm the ability of the ciPTEC-OAT1 to handle uremic toxins, cells were co-incubated with fluorescein and PBUTs (indoxyl sulfate, IS 110 µM; kynurenic acid, KA 1 µM; p-cresylsulfate, PCS 125 µM), following the procedure described above. In order to replicate the uremic condition present in CKD patients, a specific mixture of eight known anionic PBUTs (Appendix A), predominantly derived from endogenous metabolism pathways and food digestion in the gut, was used at concentrations corresponding to those found in patients. The inhibition of OAT1-mediated fluorescein uptake during the co-exposure to drugs (variable concentrations) and PBUTs (fixed concentration) was determined by the percentage reduction of fluorescein uptake in the cells. Additionally, concentration-dependent mediated fluorescein uptake (concentration range = 0–3 µM) in the presence of PBUTs (IS, KA, pCS and UTox) was studied.

### 4.5. Data Analysis

All data are expressed as mean ± standard deviation (SD) of at least three separate experiments. Inhibition data were fitted according to one-site total binding saturation curve using non-linear regression analysis ((inhibitor) vs. response-variable slope) and Vmax values were calculated according to Michaelis-Menten kinetics using non-linear regression analysis. Statistical analysis was performed using one-way ANOVA analysis followed by Tukey’s multiple comparison test (OAT1-mediated inhibition uptake of fluorescein in the presence of PBUTs, reference values = arbitrary units of fluorescence when cells are exposure to fluorescein 1 µM = 100%) or two-way ANOVA analysis followed by Dunnett’s Multiple Comparison test (drug–toxins inhibition of OAT1-mediated fluorescein uptake, reference values = fluorescein percentage in the presence of toxin alone) with GraphPad Prism version 8.4 (La Jolla, CA, USA). All experiments were performed four times.

## Figures and Tables

**Figure 1 toxins-12-00391-f001:**
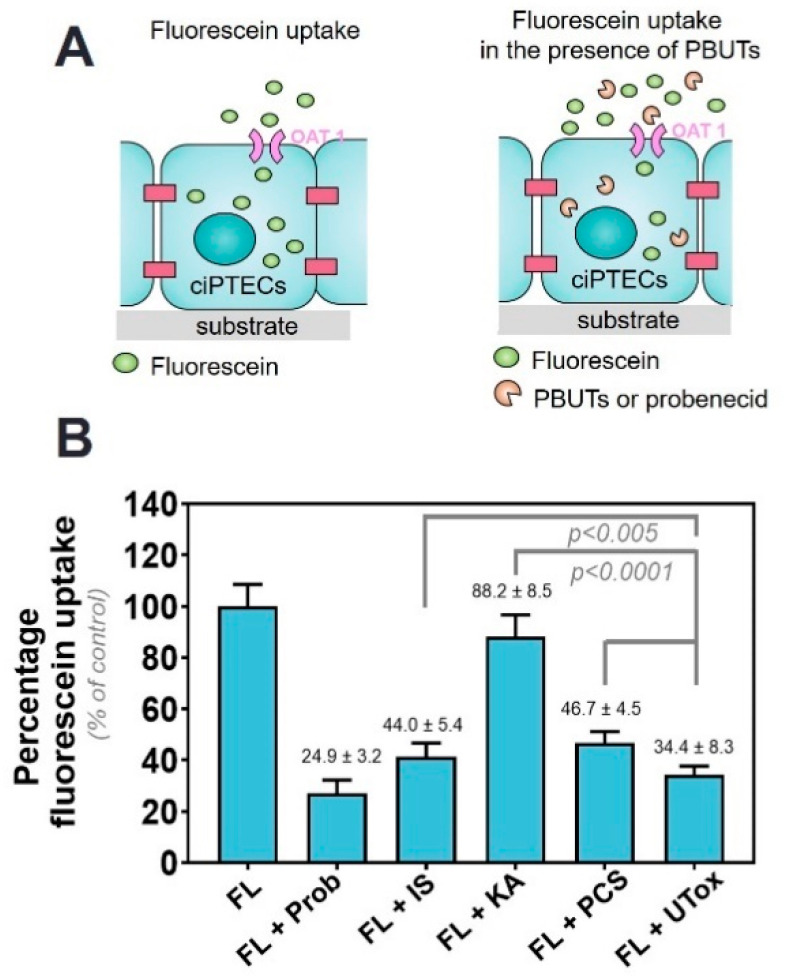
Organic anion transporter (OAT) 1-mediated uptake of fluorescein (FL) in the presence of protein-bund uremic toxins (PBUTs). (**A**) Schematic representation of the experiment. To evaluate the potency of the transporter, fluorescein was incubated with matured ciPTECs. Fluorescein uptake was then quantified and assigned as 100% uptake. Secondly, fluorescein was incubated together with either probenecid, a known inhibitor of OAT1 activity, or individual PBUTs (IS = indoxyl sulfate, KA = kynurenic acid, PCS = p-cresylsulfate) and UTox (a mix of 8 PBUTs at concentrations found in uremic serum). (**B**) The intracellular accumulation of fluorescein was measured. Data are shown as mean ± SD of four independent experiments, performed in triplicate. *p* < 0.005 and *p* < 0.0001 using one-way ANOVA analysis followed by a Tukey’s multiple comparison test.

**Figure 2 toxins-12-00391-f002:**
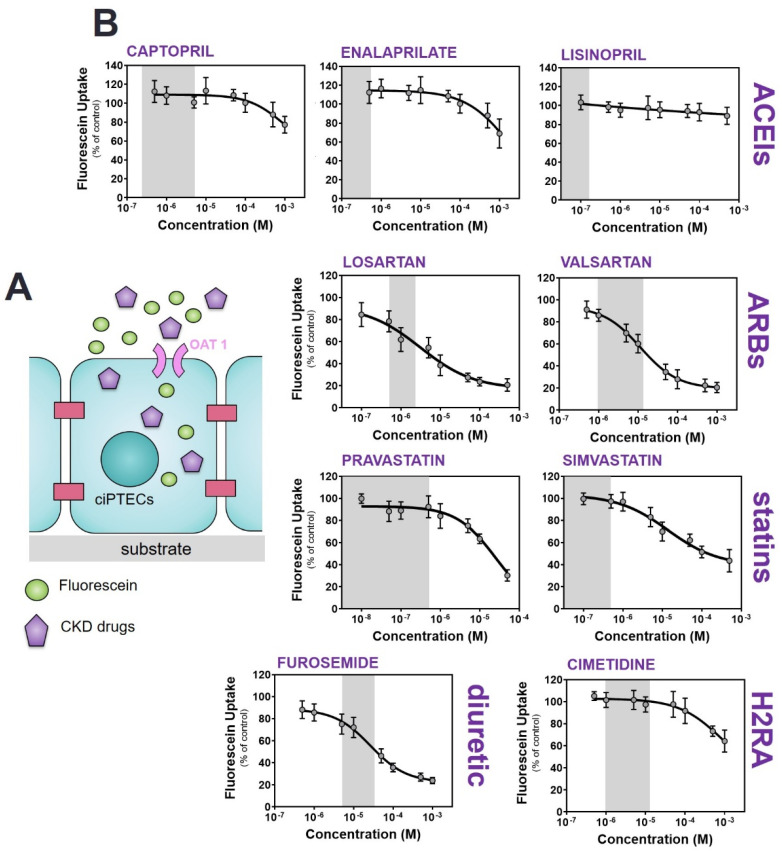
Inhibition of OAT1-mediated fluorescein uptake by a panel of drugs commonly used in CKD management. (**A**) Schematic representation of the co-incubation of fluorescein with variable concentration of drugs. (**B**) Fluorescein uptake (1 µM) by ciPTEC-OAT1 in the presence of drugs (ACEIs: captopril, enalaprilate and lisinopril; ARBs: losartan and valsartan; statins: pravastatin and simvastatin; diuretics: furosemide) relative to the uptake of fluorescein without drugs (=100%). The histamine H2 receptor antagonist (H2RA): cimetidine) was used as a reference of no inhibitory effect on OAT1-mediated fluorescein. All data are expressed as mean ± SD of four independent experiments. Grey regions indicate the therapeutic window of the respective drug.

**Figure 3 toxins-12-00391-f003:**
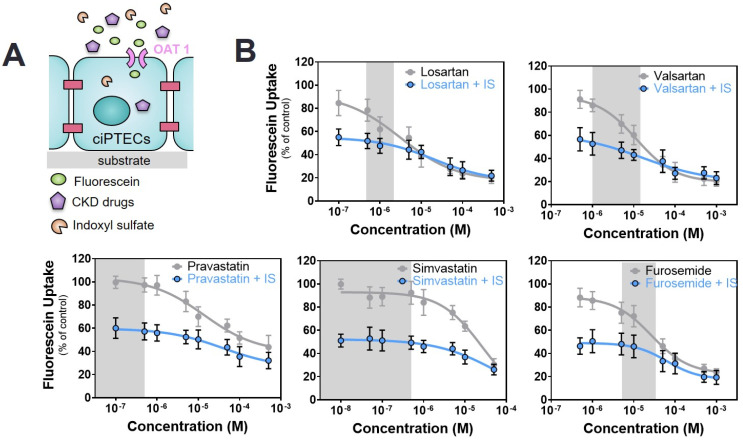
Combined interaction of drugs and indoxyl sulfate (IS) on OAT1-mediated fluorescein uptake. (**A**) Schematic representation of the co-incubation of fluorescein with variable concentrations of drugs and fixed concentration of IS (blue line). (**B**) The drug alone (grey line) shows a concentration-dependent inhibitory effect. Co-incubation with IS (110 µM, blue line) results in a further decrease of fluorescein uptake. All data are expressed as mean ± SD of four independent experiments. Curves were obtained after non-linear regression analysis. The reference value for the drug alone is fluorescein without the drug (100%), while for the co-incubation with IS the reference value is fluorescein in the presence of IS without the drug (44.0 ± 5.4%). Grey regions indicate the therapeutic window of the respective drug.

**Figure 4 toxins-12-00391-f004:**
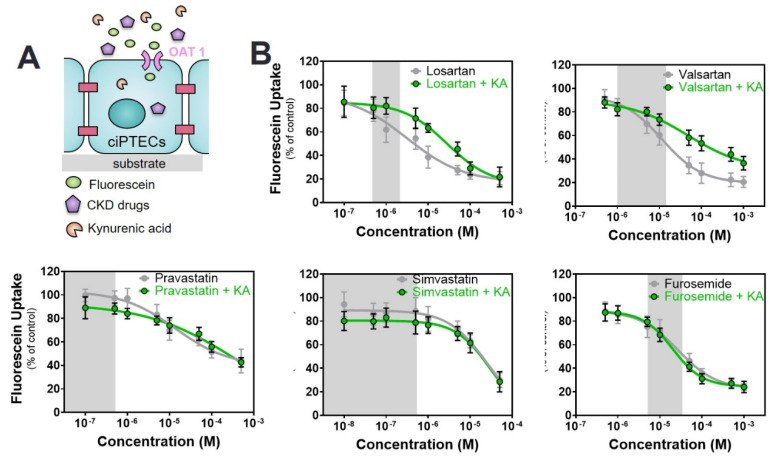
Combined interaction of drugs and kynurenic acid (KA) on OAT1-mediated fluorescein uptake. (**A**) Schematic representation of the co-incubation of fluorescein with variable concentrations of drugs and fixed concentration of KA. (**B**) The drug alone (grey line) shows a concentration-dependent inhibitory effect that is not affected by the presence of KA (1 µM, green line) in the case of furosemide and statins. For ARBs, the inhibitory effect of the drugs is diminished in the presence of KA. All data are expressed as mean ± SD of four independent experiments. Curves were obtained after non-linear regression analysis. The reference value for the drug alone is fluorescein without the drug (100%), while for the co-incubation with KA the reference value is fluorescein in the presence of KA without the drug (88.2 ± 8.5). Grey regions indicate the therapeutic window of the respective drug.

**Figure 5 toxins-12-00391-f005:**
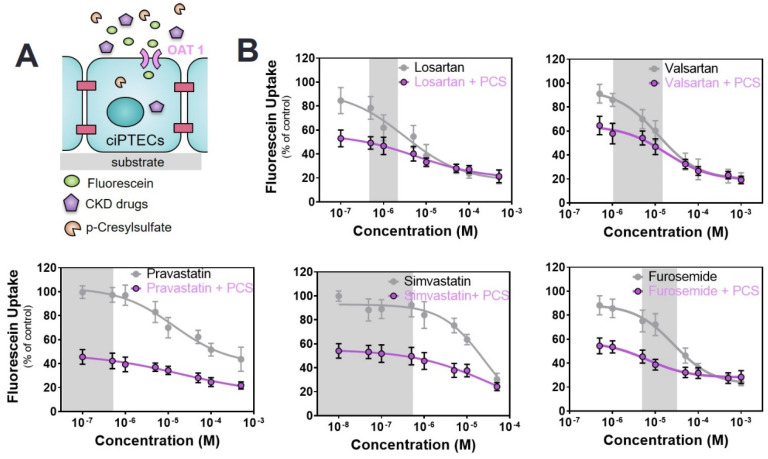
Combined interaction between drugs and p-cresylsulfate (PCS) on OAT1-mediated fluorescein uptake. (**A**) Schematic representation of the co-incubation of fluorescein with variable concentrations of drugs and fixed concentration of PCS. (**B**) The drug alone (grey line) shows a concentration-dependent inhibition effect. During the co-incubation of cells with variable drug concentrations and a fixed concentration of PCS (125 µM, purple line), the concentration-dependent inhibition trend is maintained. All data are expressed as mean ± SD of four independent experiment. Curves were obtained after non-linear regression analysis. The reference value for the drug alone (grey line) is fluorescein without the drug (100%), while for the co-incubation with PCS (purple line) the reference value is fluorescein in the presence of PCS without the drug (46.7 ± 4.5%). Grey regions indicate the therapeutic window of the respective drug.

**Table 1 toxins-12-00391-t001:** Inhibitory potencies of selected drugs on fluorescein uptake in ciPTEC-OAT1.

Drug	IC_50_ (µM) ^a^	*R* Square ^b^	Therapeutic Concentrations (µM) [44]
**ACEIs**			
Captopril	2022 ± 465	0.5557	0.2–5
Enalaprilate	1853 ± 370	0.6753	0.04–0.4
Lisinopril	not available	0.1671	0.01–0.16
**ARBs**			
Losartan	3.1 ± 0.7	0.8713	0.5–1.5
Valsartan	11.5 ± 3.5	0.9374	2–14
**Diuretics**			
Furosemide	28.1 ± 9.1	0.9301	6–30
**Statins**			
Pravastatin	13.8 ± 8.5	0.8757	0.08–0.3 [45]; 0.43 [46,47]
Simvastatin	21.3 ± 3.8	0.8613	0.006–0.014 [44]; 0.55 [46,47]
**H2RA**			
Cimetidine	887.6 ± 198.4	0.7441	1–16

^a^ Data are expressed as mean ± SD. ^b^ Curves were obtained after non-linear regression analysis. Abbreviations: ACEIs = angiotensin-converting enzyme inhibitors; ARBs = angiotensin receptor blockers; H2RA = histamine H_2_ receptor antagonists, IC_50_ = half maximal inhibitory concentration.

**Table 2 toxins-12-00391-t002:** OAT1-mediated fluorescein uptake: IC_50_ (µM) in the absence (−) and presence (+) of individual uremic toxins (IS, KA and PCS).

Drug/Toxin	−	+IS 110 µM	+KA 1 µM	+PCS 125 µM
**ARBs**				
Losartan	8.6 ± 2.5	13.9 ± 5.9 ^b^	28.2 ± 2.7 ^a^	15.97 ± 3.9 ^a^
Valsartan	11.5 ± 3.5	16.1 ± 3.6 ^b^	46.9 ± 4.6 ^a^	17.9 ± 3.8 ^b^
**Statins**				
Pravastatin	13.8 ± 8.35	40.9 ± 9.2 ^a^	243.0 ± 45.8 ^a^	19.1± 3.2 ^a^
Simvastatin	21.3 ± 3.8	71.8 ± 27.3 ^a^	28.4 ± 10.1 ^b^	32.8 ± 7.6 ^a^
**Diuretics**				
Furosemide	28.1 ± 9.1	44.7 ± 12.4 ^a^	24.8 ± 6.2 ^b^	60.2 ± 1.0 ^a^

^a^ significant increase when compared with IC_50_ values of corresponding drug alone, ^b^ no significant change when compared with IC_50_ values of corresponding drug alone. Abbreviations: IS = indoxyl sulfate; KA = kynurenic acid; PCS = p-cresylsulfate; ARBs = angiotensin receptor blockers.

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
