# Peer review of "Drugs Commonly Applied to Kidney Patients May Compromise Renal Tubular Uremic Toxins Excretion"

_toxins, 2020, doi:10.3390/toxins12060391_

Round 1

Reviewer 1 Report

The authors reported the inhibitory effect of uremic toxins on the uptake of renal tubular cells medicated by organic anion transporter. Some ARB, ACEI, furosemide, and statin were suppressed to uptake in ciPTEC with OAT-1 expression by indoxyl sulfate and p-cresyl sulfate using fluorescein assay.

This investigation may be important to understand competing of uremic toxins and drugs to the cells.

1.Most of drugs that the authors examined are known to be metabolized mainly in liver. Then, it is difficult to find the importance of this study using renal tubular cell line. They should discuss more deeply about the meaning of this study for the clinical situation.

2.Figure 1. UTox contains IS, KA and pCS. It makes us confuse to understand Fig1B right column, because it included IS, KA, pCS and UTox. The authors should revise it.

  1. Abstract. The authors should describe the actual name of ARB and ACEi. Also, it will be easy to imagine the inhibitory effect of uremic toxins if the authors put the actual data in the Abstract.

Author Response

The authors reported the inhibitory effect of uremic toxins on the uptake of renal tubular cells medicated by organic anion transporter. Some ARB, ACEI, furosemide, and statin were suppressed to uptake in ciPTEC with OAT-1 expression by indoxyl sulfate and p-cresyl sulfate using fluorescein assay. This investigation may be important to understand competing of uremic toxins and drugs to the cells.

  1. Most of drugs that the authors examined are known to be metabolized mainly in liver. Then, it is difficult to find the importance of this study using renal tubular cell line. They should discuss more deeply about the meaning of this study for the clinical situation.

We acknowledge the important and valid comment and we appreciate the opportunity to explain. Interactions with PBUTs go beyond renal clearance solely. A drug is considered to be renally cleared when 25% or more is being excreted unchanged via the kidneys (Morrissey et al. Annu Rev Pharmacol Toxicol. 2013;53:503-29). For these drugs, any effect on kidney function or interaction with renal excretion can result in pharmacodynamic changes. Indeed, some of drugs that we evaluated in this study have a limited renal clearance, whereas others are highly renally cleared. This information is now added in the supplementary section as Table S6. Of note, simvastatin shows a highly limited bioavailability (5%) and is limitedly renally cleared (13%), but shows a high protein binding (95%) that could also interact with PBUTs. In Table S6, we also added the percentage protein binding of the drugs to give more detailed information on pharmacokinetic parameters. In addition, CKD affects absorption in the gut and metabolism by the liver as well. We adapted the discussion part so that these aspects are included. Additionally, the contribution of liver clearance in understanding drug-toxin interaction in CKD and potential clinical implications are covered in the discussion.

Table S6. Physicochemical determinants of human renal clearance of tested drugs

Drug

Total CL

(mL/min/kg) [76]

Renal CL

(mL/min/kg) [76]

Renal clearance (% of total CL)

Protein binding (%)

(www.drugbank.ca)

T1/2 (h)

[44]

ACEIs

Captopril

12

12

100

25-30

1-2

Enalaprilate

1.6

1.6

100

50

8-11

Lisinopril

1.2

1.2

100

negligible

12

ARBs

Losartan

8.2

0.9

11

99

1.5-2

Valsartan

0.49

0.14

29

94-97

6-9

DIURETICS

Furosemide

2.4

1.7

71

95-99

1-3

STATINS

Pravastatin

14

6.3

45

60

1-2.5 [45]

Simvastatin

-

-

13

95

2

H2RA

Cimetidine

8.1

7.9

97

20

1.5-4

The goal of our study was to assess if commonly prescribed drugs could interfere with the clearance of PBUTs, focusing on renal clearance solely. Since the first step in PBUTs excretion is the uptake by the proximal tubule cells via OAT1 activity, we considered that potential PBUTs-drug interactions should be first evaluated in a proximal tubule-like OAT1-expressing cell line (ciPTECs-OAT1). Interestingly, our study shows that even drugs that are not predominantly eliminated via kidney, can inhibit the OAT1-mediated fluorescein uptake, indicating an interaction with the transporter. Some of these drugs were previously reported to potentially inhibit OATs activity (valsartan inhibits OAT1: Huo X, J Pharm Sci. 2014; 103(2), simvastatin inhibits OAT1: Windass AS, J Pharmacol Exp Ther. 2007;322(3), pravastatin inhibits OAT3: Takeda M, Eur J Pharmacol. 2004;483(2-3), losartan inhibits OAT1 Race JE, Biochem Biophys Res Commun. 1999;255(2): and inhibits OAT4 Yamashita F, J Pharm Pharmacol. 2006;58(11)), but not in the context of CKD, and not to mention the interactions with PBUTs.

  1. Figure 1. UTox contains IS, KA and pCS. It makes us confuse to understand Fig1B right column, because it included IS, KA, pCS and UTox. The authors should revise it.

The figure has been revised to provide a better clarity in the description of the conditions.

              Revised Figure 1. Organic anion transporter (OAT)1-mediated uptake of fluorescein (FL) in the presence of protein-bund uremic toxins (PBUTs). (A) Schematic representation of the experiment. To evaluate the potency of the transporter, FL was incubated with matured ciPTECs. FL uptake was then quantified and assigned as 100% uptake. Secondly, FL was incubated together with either probenecid, a known inhibitor of OAT1 activity, or individual PBUTs (IS = indoxyl sulfate, KA = kynurenic acid, PCS = p-cresylsulfate) and UTox (a mix of 8 PBUTs at concentrations found in uremic serum). (B) The intracellular accumulation of FL was measured. Data are shown as mean ± SD of four independent experiments, performed in triplicate. P<0.005 and p<0.0001 using one-way ANOVA analysis followed by a Tukey’s multiple comparison test.

  1. The authors should describe the actual name of ARB and ACEi. Also, it will be easy to imagine the inhibitory effect of uremic toxins if the authors put the actual data in the Abstract.

We acknowledge the comment and updated the abstract accordingly.

Reviewer 2 Report

Reviewer comments:

In this study the authors investigate the interaction between commonly prescribed drugs in CKD and endogenous protein-bound uremic toxins with respect to OAT1-mediated uptake. In an OAT1-mediated fluorescein assay in proximal tubule cell line (ciPTEC), authors found that exposed cells to a selected ARBs and furosemide significantly reduced fluorescein uptake. This was exaggerated in presence of some PBUTs, while selected ACEIs and statins had either no or a slight effect at supratherapeutic concentrations. While the study is interesting, well conducted and addresses an important clinical question, in its current form, the manuscript requires revision before it will be suitable for publication.

Major points:

1/

In the abstract section, the authors must write:

…. “This was exaggerated in presence of SOME PBUTs”.

2/

In results presented in Figure 3 and 5, for some drugs authors found fluorescein uptake showed a decrease with increasing drug at therapeutic concentrations, when co-incubated with Is or PCS. Are these differences significative? This point must be analysed.

3/

The authors should consider the possibility of discussing the last result mentioned:

“The exposure to the mixture of PBUTs (UTox) resulted in a complete reduction in fluorescein uptake, with no additional effect of the drugs”

4/

Figure 1:

What exactly is the composition of the UTox?

According to legends of figure 1B, “UTox (a mix of 8 PBUTs at concentrations found in uremic serum)”

This is not exact if we take into account the data in Table 1 of the supplementary material, especially for the p-Cresylsulfate

5/

Figures legends:

Where applicable, in the footnotes of the corresponding figures, explained what represents the gray shaded area.

Minor points:

6/

Supplementary material,

Fig 2 text say

…”The incubation with UTox alone results in strong inhibitory effect…(this is data not show??)

7/

Supplementary material,

Why are different toxin values ​​used when treated individually or in the UTox mixture, as indicated by the values ​​in the third column at the bottom of Table 1?

8/

The next paragraphs must be rewritten in order to correct the grammatical errors

- Materials and Methods: line 321, should say: 490 nm

Author Response

In this study the authors investigate the interaction between commonly prescribed drugs in CKD and endogenous protein-bound uremic toxins with respect to OAT1-mediated uptake. In an OAT1-mediated fluorescein assay in proximal tubule cell line (ciPTECs), authors found that exposed cells to a selected ARBs and furosemide significantly reduced fluorescein uptake. This was exaggerated in presence of some PBUTs, while selected ACEIs and statins had either no or a slight effect at supratherapeutic concentrations. While the study is interesting, well conducted and addresses an important clinical question, in its current form, the manuscript requires revision before it will be suitable for publication.

Major points:

  1. In the abstract section, the authors must write: “This was exaggerated in presence of SOME PBUTs”.

We have modified the text accordingly.

  1. In results presented in Figure 3 and 5, for some drugs authors found fluorescein uptake showed a decrease with increasing drug at therapeutic concentrations, when co-incubated with Is or PCS. Are these differences significative? This point must be analysed.

We acknowledge this comment. We have included a supplementary table (Table S5) which depicts the drug concentration at which the percentage of fluorescein uptake is significantly decreased when compared to PBUTs alone. In the discussion part, we refer to this table when discussing about the simultaneous incubation of PBUTs with the selected drugs, that resulted in significant alteration of fluorescein uptake. We emphasize that alterations occur at concentrations that fall in the therapeutic window of the drugs.  

Table S5. Overview of the lowest drug concentrations at which a statistically significant decrease (p<0.05) of the percentage (%) of fluorescein uptake was observed

Drug concentration (µM)

No Drug

ARBs

STATINS

DIURETICS

PBUTs

% FL uptake

reference values

Losartan

Valsartan

Pravastatin

Simvastatin

Furosemide

IS

44.0 ± 5.4

1*

5*

5

5

50

KA

88.2 ± 8.5

5

10*

5

10

5*

PCS

46.7 ± 4.5

1*

1*

1*

1*

5*

*indicates concentrations that fall in the therapeutic concentration range 

  1. The authors should consider the possibility of discussing the last result mentioned: “The exposure to the mixture of PBUTs (UTox) resulted in a complete reduction in fluorescein uptake, with no additional effect of the drugs”

We added a statement in the results section that addresses the suggestion of the reviewer and better reflects the significance of the results: ”The exposure to the mixture of PBUTs (UTox) resulted in a strong reduction of fluorescein uptake (34.4 ± 8.3%). With the addition of drugs, a further decrease was observed at the lowest concentration applied (with values comparable to probenecid), with a more pronounced effect for losartan and valsartan. This drop remained stable with the increase of the drug concentration (Figure S2, Table S4).”

We have also addressed this issue in the discussion part with the following statement: “When co-incubating with selected drugs, a further decrease in fluorescein uptake was observed, reaching similar levels as those obtained with probenecid, suggesting a complete saturation of the transporter via competitive inhibition”

Table S4. Overview of the percentage (%) reduction of fluorescein uptake in the presence of UTox and drugs (at the lowest concentration)

Percentage (%) fluorescein uptake at the lowest drug concentration

No Drug

ARBs

STATINS

    DIURETICS

reference

Losartan

Valsartan

Pravastatin

Simvastatin

Furosemide

UTox

34.4 ± 8.3

23.5 ± 3.5

24.7 ± 4.5

27.8 ± 2.6

25.7 ± 2.9

28.7 ± 3.2

  1. Figure 1: What exactly is the composition of the UTox? According to legends of figure 1B, “UTox (a mix of 8 PBUTs at concentrations found in uremic serum)” This is not exact if we take into account the data in Table 1 of the supplementary material, especially for the p-Cresylsulfate and Indole-3-acetic acid, whose concentration are reflecting the uremic conditions (125 uM and 10uM respectively)

We have used the uremic toxin mix recipe developed and previously published (Mihajlovic M, Int J Mol Sci. 2017;18(12):2531, doi:10.3390/ijms18122531), with minor adjustments that were not properly depicted in Table 1 of the supplementary section. We would like to confirm that we have used a modified composition of the previously established UTox, viz. the concentrations of PCS (125 µM), KA (1µM) and of Indole-3-acetic acid (10µM) to better reflect CKD plasma composition. This revision is now added to the methods section of our revised manuscript.

  1. Figures legends: Where applicable, in the footnotes of the corresponding figures, explained what represents the gray shaded area.

We have included the explanatory note (“Grey regions indicate the therapeutic window of the respective drug.”) in Figure 3, 4 and 5 and Figure S2.

Minor points:

  1. Supplementary material, Fig 2 text say ”The incubation with UTox alone results in strong inhibitory effect…(this is data not show??)

The results of the OAT1-mediated uptake of fluorescein in the presence of UTox (alone) are described in Figure 1, where we show a significant decrease in fluorescein uptake (34.4 ± 8.3 %).

  1. Supplementary material: Why are different toxin values ​​used when treated individually or in the UTox mixture, as indicated by the values ​​in the third column at the bottom of Table 1?

As mentioned in our response to a previous comment, the composition of UTox mixture was not properly depicted in supplementary Table 1. The concentrations of individual IS, PCS and KA are reflected in the composition of UTox mixture. 

  1. The next paragraphs must be rewritten in order to correct the grammatical errors

- Materials and Methods: line 321, should say: 490 nm

We have modified the text accordingly.

Round 2

Reviewer 2 Report

No coments